# Relationship of Extravascular Lung Water and Pulmonary Vascular Permeability to Respiratory Mechanics in Patients with COVID-19-Induced ARDS

**DOI:** 10.3390/jcm12052028

**Published:** 2023-03-03

**Authors:** Florian Lardet, Xavier Monnet, Jean-Louis Teboul, Rui Shi, Christopher Lai, Quentin Fossé, Francesca Moretto, Thibaut Gobé, Ludwik Jelinski, Margot Combet, Arthur Pavot, Laurent Guérin, Tài Pham

**Affiliations:** 1Service de Médecine Intensive-Réanimation, Hôpital de Bicêtre, DMU CORREVE, FHU SEPSIS, Groupe de Recherche CARMAS, Hôpitaux Universitaires Paris-Saclay, AP-HP, 94270 Le Kremlin-Bicêtre, France; 2Département d’Anesthésie-Réanimation et Médecine Périopératoire, Hôpital Tenon, AP-HP, Sorbonne Université, 75020 Paris, France; 3INSERM UMR S_999 “Pulmonary Hypertension: Pathophysiology and Novel Therapies”, Hôpital Marie Lannelongue, 92350 Le Plessis-Robinson, France; 4INSERM U1018, Equipe d’Epidémiologie Respiratoire Intégrative, CESP, Université Paris-Saclay (UVSQ)—Université Paris-Sud, 94807 Villejuif, France

**Keywords:** SARS-CoV-2, acute respiratory distress syndrome, transpulmonary thermodilution, mechanical ventilation, lung compliance, driving pressure

## Abstract

During acute respiratory distress syndrome (ARDS), the increase in pulmonary vascular permeability and lung water induced by pulmonary inflammation may be related to altered lung compliance. A better understanding of the interactions between respiratory mechanics variables and lung water or capillary permeability would allow a more personalized monitoring and adaptation of therapies for patients with ARDS. Therefore, our main objective was to investigate the relationship between extravascular lung water (EVLW) and/or pulmonary vascular permeability index (PVPI) and respiratory mechanic variables in patients with COVID-19-induced ARDS. This is a retrospective observational study from prospectively collected data in a cohort of 107 critically ill patients with COVID-19-induced ARDS from March 2020 to May 2021. We analyzed relationships between variables using repeated measurements correlations. We found no clinically relevant correlations between EVLW and the respiratory mechanics variables (driving pressure (correlation coefficient [CI 95%]: 0.017 [−0.064; 0.098]), plateau pressure (0.123 [0.043; 0.202]), respiratory system compliance (−0.003 [−0.084; 0.079]) or positive end-expiratory pressure (0.203 [0.126; 0.278])). Similarly, there were no relevant correlations between PVPI and these same respiratory mechanics variables (0.051 [−0.131; 0.035], 0.059 [−0.022; 0.140], 0.072 [−0.090; 0.153] and 0.22 [0.141; 0.293], respectively). In a cohort of patients with COVID-19-induced ARDS, EVLW and PVPI values are independent from respiratory system compliance and driving pressure. Optimal monitoring of these patients should combine both respiratory and TPTD variables.

## 1. Introduction

During acute respiratory distress syndrome (ARDS), lung inflammation increases pulmonary vascular permeability (PVP), leading to extravascular lung water (EVLW) accumulation [1]. The increase in EVLW, which can be considered as the pathophysiological hallmark of ARDS [2], is related to the degree of diffuse alveolar damage [3] and to the lung weight [4]. The EVLW and PVP can be assessed at the bedside by transpulmonary thermodilution (TPTD) [5], and both EVLW indexed for body weight (EVLWi) and PVP index (PVPI) estimated by this technique have been shown to be related to outcome independently from other factors, including in ARDS patients [6].

The treatment of ARDS is based on mechanical ventilation [7,8]. Lung mechanics-related variables, especially the airway driving pressure, impact the outcome of patients with ARDS, again independently from other markers of severity [9].

The accumulation of EVLWi on the one hand and the impairment of lung mechanics on the other hand may be linked in patients with ARDS treated with invasive mechanical ventilation. Indeed, there may be a relationship between the increase in lung weight and the loss of lung aeration indicated by EVLWi and the decrease in lung compliance [5,7,10,11]. However, other factors could contribute to this decrease, such as pulmonary consolidation and atelectasis or lung fibrosis at late stages of the disease [12]. If this relationship between EVLWi accumulation and the impairment of lung compliance exists, it could impact patients’ monitoring and management. For instance, it would be a strong rationale for studies investigating the impact of fluid therapy on lung mechanics. Restricting fluid administration in ARDS reduces EVLWi accumulation [13]. It has also been shown that fluid restriction improves lung function and reduces the duration of mechanical ventilation [14]. However, whether the fluid management can influence the lung compliance and driving pressure has not been clearly proven. Showing a relationship between EVLWi and the lung mechanical properties would open the possibility to investigate this issue.

Few studies have assessed the association between EVLWi or PVPI and mechanical ventilation variables [15,16,17,18,19,20]. They found discrepant results, with either no correlation [15,16,18] or an inverse correlation between EVLWi or PVPI and lung compliance [17,19,20]. However, all these studies had small sample sizes and included heterogeneous populations of subjects. Moreover, in all these studies, the assessment of the link between EVLWi and respiratory mechanics variables was not the main goal.

Our primary objective was to assess the relationship between EVLWi or PVPI on the one side and respiratory system compliance or driving pressure on the other side in a homogeneous population of patients with ARDS induced by Coronavirus Disease 2019 (COVID-19).

## 2. Materials and Methods

### 2.1. Study Design and Patients

This observational retrospective cohort study was performed in the medical intensive care unit (ICU) of the Bicêtre hospital (Assistance Publique—Hôpitaux de Paris) between March 2020 and May 2021. We screened all adult patients admitted in this unit, who presented the following inclusion criteria: (i) diagnosis of ARDS [21], (ii) invasive mechanical ventilation, (iii) COVID-19 diagnosed by polymerase chain reaction and recognized as the main cause for ARDS, (iv) already monitored with a TPTD device (PICCO2, Pulsion Medical Systems, Getinge, Feldkirchen, Germany) and (v) at least one concomitant measurement of EVLWi or PVPI and respiratory mechanics (total positive end-expiratory pressure (PEEP), plateau pressure (Pplat)).

Exclusion criteria were (i) treatment with extracorporeal membrane oxygenation, as the estimation of EVLWi and PVPI by TPTD is not reliable under this treatment [22], and (ii) impossibility to perform respiratory mechanics measurements due to technical reasons.

The study was a part of the “Cohort study for identifying Criteria of Treatment Individualization in Patients with Sepsis” (COTIPS study), which has been approved by the Ethics Committee of the French Intensive Care Society (CE SRLF 20-82). All data were prospectively collected through the latter study, but the present ancillary analysis was designed after data collection. Patients, or their surrogate decision makers, received appropriate information, in accordance with French law.

### 2.2. Data Collection

Demographic and physiological data, medical history, hemodynamic and respiratory variables and outcomes were collected through the patients’ medical chart (paper and electronic charts). For TPTD measurements, boluses of cold saline were injected through a central venous catheter inserted in the internal jugular vein and a thermodilution curve was recorded by a thermistor-tipped catheter inserted through the femoral artery [23]. The mean of three consecutive measurements was used for TPTD-derived variables [24]. Patients received invasive mechanical ventilation in volume assist-control mode. As per usual care in our ICU, tidal volume was set at 6 mL/kg of predicted body weight. The level of PEEP was set according to the decision of clinicians in charge but always keeping Pplat ≤ 30 cmH_2_O. Respiratory mechanics were assessed while the patient was passively ventilated (no triggering or effort was observed on the airway pressure curve or the flow curve of the ventilator screen). For this purpose, neuromuscular blocking agents could be administered. Pplat was measured during a 3-s end-inspiratory occlusion and total PEEP was measured during a 3-s end-expiratory pause. Driving pressure was calculated as the difference between the Pplat and total PEEP. Compliance of the respiratory system (Crs) was calculated as tidal volume divided by the driving pressure.

The “minimum” and “maximum” values of variables were defined as the lowest and highest values, respectively, among all the measurements performed in each patient while the TPTD device was in place. In order to assess these values in the most acute phase, we also collected these “minimum” and “maximum” values restricted to the first week following intubation.

Frailty was defined as a score of five or more on the clinical frailty scale, which corresponds to patients who are mildly, moderately, severely, very severely frail or terminally ill [25]. Ventilator-free days were calculated as the number of days from weaning from invasive ventilation to day 28. Patients who died still receiving invasive ventilation were considered to have a ventilator-free-day value of 0. Respiratory mechanics and TPTD variables were assessed daily as part of usual care.

### 2.3. Data Quality Control

Prior to analysis, all data were screened for potentially erroneous data and outliers and these data were verified and corrected. Outlier data were carefully searched and checked to be confirmed or corrected. We followed the STROBE (Strengthening The Reporting of OBservational studies in Epidemiology) statement guidelines for observational cohort studies [26] (see Appendix A).

### 2.4. Statistical Analysis

Continuous variables are reported as mean (standard deviation) (SD) or median [1st; 3rd quartiles] and categorical variables as count and proportion. Normality of the data distribution for continuous data was visually assessed by means of histograms and these variables were compared using Student *t*-test or Wilcoxon rank sum test, as appropriate. Proportions were compared using chi-square or Fisher exact tests.

As collection of respiratory and TPTD variables was repeated over time, different measurements from the same patients were not independent and we assessed association using repeated measures correlation techniques [27,28,29]. Correlations coefficient absolute value above 0.5 and 0.7 were considered as “moderate” and “strong” correlations, respectively. We performed bivariate analyses to assess factors associated with ICU mortality.

No statistical power calculation was performed before the study, and sample size was based on available data. No assumptions were made for missing data. Statistical analyzes were performed with R 4.21 (R Foundation for Statistical Computing, Vienna, Austria, http://www.R-project.org). All *p* values were two-sided, and values less than 0.05 were deemed statistically significant.

## 3. Results

### 3.1. Patient Characteristics and Outcome

During the inclusion period, a total of 956 patients were admitted in our unit; 430 received invasive mechanical ventilation and 246 of them presented ARDS related to COVID-19 (Figure 1). Among those, 107 were included in this study: 37 (35%) patients were included during the first wave of the pandemic in France (from March 2020 to August 2020) and 70 (65%) during the second wave (September 2020 to May 2021). Most patients were male (78%) and their mean (SD) age was 64 (11) years. Eighty (75%) had at least one significant comorbidity, most frequently hypertension (50%, *n* = 53), diabetes mellitus (32%, *n* = 34) and obesity (31%, *n* = 33) (Table 1). On admission, mean (SD) of Sepsis-related Organ Failure Assessment (SOFA) and simplified acute physiology score II (SAPSII) scores were 5 (3) and 39 (15), respectively. Patients were intubated 1 (0; 3) day after the admission in the ICU. The median (IQR) duration of mechanical ventilation was 18 (9; 29) days and the median (IQR) number of ventilator free days was 0 (0; 4) days. The median (IQR) length of stay in the ICU was 19 (11; 32) days. A total of 62 patients (58%) died in the ICU. No right severe tricuspid regurgitation was detected in these patients through echocardiography. Inhaled nitric oxide was used in none of the patients.

### 3.2. Evolution of Variables with Time

On the first day of invasive mechanical ventilation, median (IQR) values of the main variables of interest were the following: driving pressure of 12 (11;16) cmH_2_O; Pplat of 27 (25;32) cmH_2_O; Crs of 32 (25;38) mL/cmH_2_O, EVLWi of 18 (15;22) mL/kg and PVPI of 3.6 (3.1;4.5) (Table 2). The evolution of EVLWi, PVPI and the driving pressure with time is shown on Figure 2. The maximal value of EVLWi and PVPI were reached 4 (1;6) days and 4 (2;7) days after intubation, respectively. The maximal value of the driving pressure was reached 5 (2;9) days after intubation. The maximal and minimal values of EVLWi, PVPI and driving pressure are provided in Table 2.

### 3.3. Correlation between EVLWi or PVPI and Respiratory Mechanics Variables

A median of 5 (2;9) measurements were performed per patient, totaling 736 assessments, of which 576 (78%) were performed in supine semi-recumbent position and 160 (22%) in prone position. These measurements showed median (IQR) driving pressure of 14 (12;17) cmH_2_O, Pplat of 28 (26;30) cmH_2_O, Crs of 29 (23;46) mL/cmH_2_O, EVLWi of 18 (15;22) mL/kg and PVPI of 3.5 (2.8;4.4).

Multiple measures correlation coefficients considered had several assessments are shown in Table 3. Though several variables showed statistically significant associations, not all were strong associations. Strong and relevant correlations were found between some respiratory mechanics variables (Pplat and Crs, Pplat and driving pressure, Crs and driving pressure), and between PVPI and EVLWi. There was no relevant correlation between EVLWi and respiratory mechanics variables (Pplat, PEEP, Crs and driving pressure) (Figure 3). Similarly, there was no relevant correlation between PVPI and respiratory mechanics variables (Pplat, PEEP, Crs and driving pressure). Considering the pairs of measurements performed on the first day of mechanical ventilation or the worst values in the first week, the correlations between EVLWi or PVPI and Pplat or Crs were also not relevant (see Appendix A, Appendix A).

## 4. Discussion

In a homogeneous population of patients receiving invasive mechanical ventilation for COVID-19-induced ARDS, we found no clinically relevant correlation between EVLWi or PVPI and respiratory mechanics variables (Pplat, driving pressure, Crs). This lack of correlation was valid when considering all pairs of measurements performed at the same time, as well as when considering only pairs of measurements performed on a given day.

It was expected to find a strong correlation between EVLW (a quantitative marker of lung edema) and PVPI (a quantitative marker of lung permeability). Patients with ARDS consistently present lung inflammation and the increase in PVPI during this syndrome might be directly linked to the extent of inflammation [2]. The EVLWi includes fluid accumulated in the interstitial and alveolar spaces and the volume of cells present in this compartment. There is a strong relationship between the value of EVLWi and the degree of diffuse alveolar damage [3]. The accumulation of EVLWi increases lung weight and may partly explain the decrease in lung compliance through mechanical changes in the pulmonary interstitium, collapse and alveolar consolidation [30,31]. The EVLWi estimated by TPTD may be influenced by the fluid strategy in ARDS patients [13] and restricting fluid administration in ARDS has been shown to improve lung function and reduce the duration of mechanical ventilation [14]. However, it has not been clearly demonstrated that fluid management can influence ventilatory mechanics. Thus, a significant relationship between EVLWi or PVPI and respiratory mechanics in mechanically ventilated ARDS patients may have some consequences and may partly explain the better ventilatory mechanics and shorter duration of ventilation in patients with restrictive fluid management. Twenty-two percent of our measurements was collected while patients were in prone position. Prone positioning could impact both EVLWi and respiratory mechanics, by increasing central venous pressure and therefore impairing pulmonary edema resorption. However, prone positioning may improve ventilatory mechanics and oxygenation, decreasing hypoxemic pulmonary vasoconstriction and improving ventilatory mechanics. The final effect is still unclear. As most measurements were made in the supine position in our study, we could not specifically investigate the effect of prone positioning on the relationship between EVLWi or PVPI and respiratory mechanics.

We did not find significant relationship between EVLWi or PVPI and respiratory mechanics. Several hypotheses may explain this result. First, the decrease in lung compliance and the increase in driving pressure during ARDS might have determinants different from the diffuse alveolar damage reflected by EVLWi. As a matter of fact, not only tissue edema, but also lung condensation, atelectasis and, at later stages, fibrosis, likely contribute to the impairment of lung mechanics [32]. Additionally, inflammation may cause structural changes, leading to a decrease in tissue elastance independently from accumulation of edema. The same above-mentioned reasons may explain that we found no relationship between PVPI and the respiratory mechanics variables. These results are in contradiction with a previous study investigating the relationship between EVLWi and respiratory mechanics as a secondary goal, which found a negative correlation between EVLWi, PVPI and Crs [17]. However, this correlation reported by Kuzkov et al. was moderate [17].

Our results suggest firstly that EVLWi and PVPI on the one hand and respiratory mechanics variables on the other provide different information and that elevated EVLW does not appear to contribute significantly to elevated ventilation pressures. However, our study confirms that both Crs and driving pressure on the one side, and EVLW and PVPI were worse in non-survivors than in survivors, suggesting the importance of these features of ARDS. Thus, these variables should be monitored independently in patients with ARDS. EVLW or PVPI are good markers of the risk of excessive fluid perfusion [5,23], while monitoring of respiratory mechanics variables is necessary to limit mechanical ventilation-induced injury. Second, our results suggest that respiratory mechanics might not be very sensitive to “restrictive” or “conservative” fluid management strategies, which may decrease the amount of lung edema without significantly improving lung compliance. Indeed, the improvement in outcomes brought about by a “restrictive” strategy in ARDS patients discussed earlier does not appear to be through an improvement in respiratory mechanics. However, Wiedemann et al. have observed that a conservative fluid management significantly reduced Pplat compared to a more liberal one [14], which may suggest an influence of fluid management strategy on respiratory mechanics. However, in this study, the group of patients with conservative fluid management had a significantly lower PEEP level. In addition, motor pressure and lung compliance were not compared between these two groups [14]. Thus, a direct investigation of this hypothesis is required.

Our cohort was a homogeneous population of patient with ARDS from a single etiology. Though major publications on dexamethasone and tocilizumab modified standard of care of critically ill patients with COVID-19 during the pandemic, most of our patients were admitted before widespread use of these treatments [33,34]. Consistent with other cohorts of critically ill patients with COVID-ARDS, most were male with relatively low severity scores (SOFA, SAPSII) on admission and their most frequent comorbidities were hypertension, diabetes and overweight [35,36,37]. These patients usually initially presented with isolated respiratory failure explaining lower admission severity scores, as described elsewhere [38,39]. This contrasts with populations of patients admitted to the ICU with ARDS of other etiology usually presenting concomitant renal, neurological and/or hemodynamic failure, resulting in higher severity score. Nevertheless, these severe patients usually presented delayed extra-pulmonary organ failures and their mortality was similar to patients with non-COVID-19 ARDS [40]. Our results on the interactions between ventilatory mechanics and TPTD measurements over time will have to be confirmed in a larger cohort of patients with ARDS of extra-pulmonary and non-COVID-19 pulmonary causes.

Our study has some limitations. First, we found a higher mortality in our cohort than in previous cohorts of patients with COVID-19 lung disease in the ICU [35,36,37]. The severity of our patients can be explained by the inclusion criteria, which selected the most severe patients. Indeed, the patients were all under invasive mechanical ventilation, treated with deep sedation and neuromuscular blocking agents for ventilatory mechanics parameters to be measured and sufficiently severe to warrant TPTD monitoring. Second, the homogeneity of the cause of ARDS could limit the generalizability of our results to other etiologies of ARDS. In a recent study comparing patients with COVID-19-induced ARDS to patients with ARDS from other causes, our team showed that ventilatory mechanics were similar but EVLW and PVPI were higher in the patients with COVID-19 ARDS [40], likely due to a higher inflammatory state. Nevertheless, this homogeneity can also be considered as a strength of our study by overcoming the large variability that can be encountered in patients with ARDS of different etiologies. Third, we did not use esophageal pressure measurements. As transpulmonary pressures differs from airway pressures, we did not measure lung compliance but only Crs [41]. This might be important in patients whose chest wall compliance is decreased and further studies are required to investigate the correlation between pulmonary driving pressures and EVLWi or PVPI. Fourth, we did not specifically investigate the role of prone position, inhaled nitric oxide and neuromuscular blocking agents. Additionally, ventilatory settings were set to limit the plateau pressure, rather than the driving pressure. This may have altered our results. Finally, we did not collect fluid balance, which is associated with outcomes of patients with ARDS [42]. Daily fluid balance is quite challenging to accurately collect in general in critically ill patients, and this was even more difficult during the increased workload due to the COVID pandemic. Positive fluid balance could increase EVLWi and worsen respiratory mechanics variables, but it might not impact our main objective that was to assess the relationship between them.

## 5. Conclusions

In this cohort of 107 patients with COVID-19-related ARDS and using more than 700 time-points collection, EVLWi and PVPI seemed to have no direct correlation with respiratory mechanics variables (PEEP, driving pressure, Pplat or Crs). Both of them provide different information and thus it is important to monitor them independently. However, our results require confirmation in a larger cohort of patients from several centers with other causes of ARDS.

## Figures and Tables

**Figure 1 jcm-12-02028-f001:**
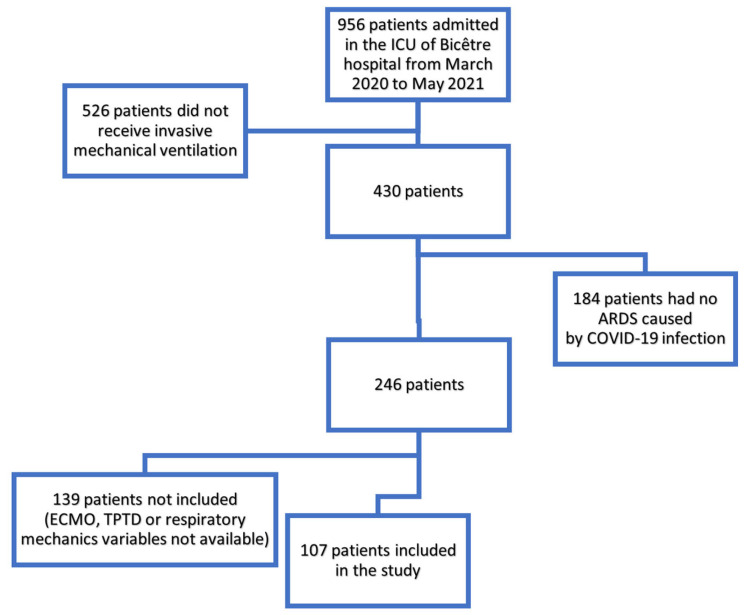
Flowchart. ARDS: acute respiratory distress syndrome; ECMO: extracorporeal membrane oxygenation; ICU: Intensive Care Unit; TPTD: transpulmonary thermodilution.

**Figure 2 jcm-12-02028-f002:**
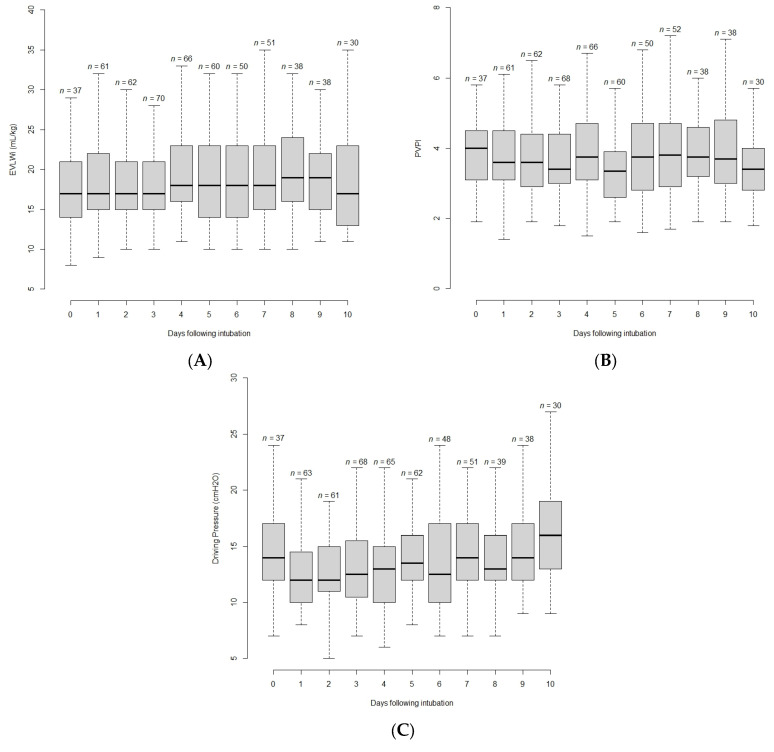
Boxplot showing time evolution of extravascular lung water indexed for ideal body weight (EVLWi) (**A**), pulmonary vascular permeability index (PVPI) (**B**) and driving pressure (**C**) from the day of intubation to the 10th day.

**Figure 3 jcm-12-02028-f003:**
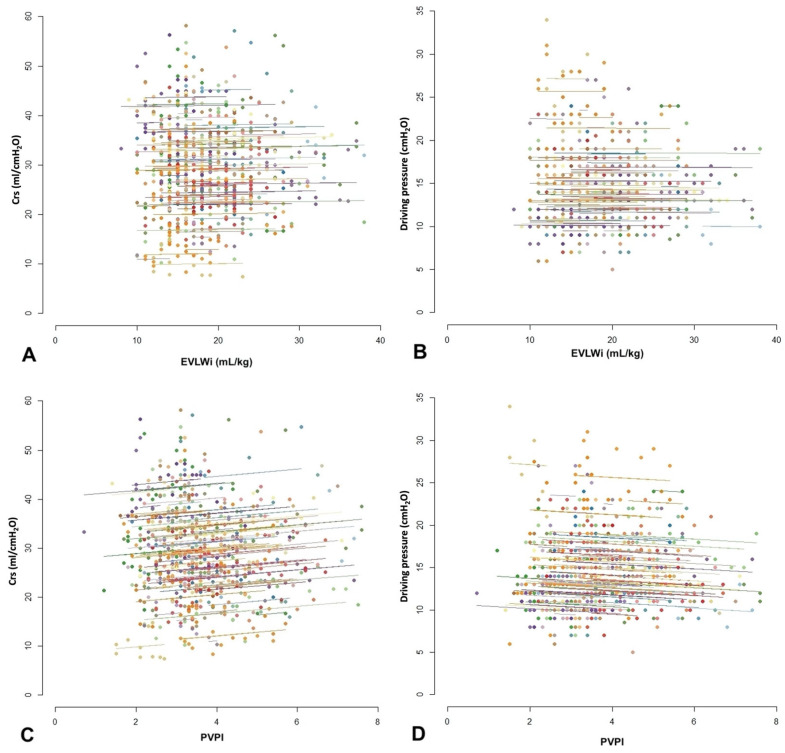
Repeated measures correlation between extravascular lung water indexed for ideal body weight (EVLWi) and compliance of the respiratory system (Crs) (**A**), EVLWi and driving pressure (**B**), pulmonary vascular permeability index (PVPI) and Crs (**C**) and between PVPI and driving pressure (**D**). The data of each participant and the corresponding regression line are shown in a different color. Multiple points of the same color represent different measurements performed on the same patient. We estimated the common regression slope for all points in the diagram, and the parallel regression lines are fitted to each participant’s data according to their corresponding color.

**Table 1 jcm-12-02028-t001:** Demographic characteristics in survivors and non-survivors.

Variables	All Patients *n* = 107 Mean ± SD or *n* (%)	Non-Survivors *n* = 62 Mean ± SD or *n* (%)	Survivors *n* = 45 Mean ± SD or *n* (%)	*p*-Value	*n*
Age, years	64 ± 11	64 ± 11	64 ± 12	0.993	107
Male	82 (76.6%)	50 (80.6%)	32 (71.1%)	0.358	107
Height, cm	171 ± 9	171 ± 9	171 ± 10	0.981	107
Weight, kg	84.0 ± 15.1	83.5 ± 15.3	84.6 ± 15.0	0.714	105
BMI, kg/m^2^	28.6 ± 4.9	28.5 ± 5.3	28.7 ± 4.4	0.831	105
Obesity, BMI > 30 kg/m^2^	33 (30.8%)	19 (30.6%)	14 (31.1%)	1.000	105
SOFA Score	5.3 ± 3.2	5.5 ± 3.5	5.2 ± 2.7	0.425	107
SAPS II Score	38.6 ± 15.1	40.6 ± 16.6	36.0 ± 12.6	0.113	105
Frailty score	4 (3.7%)	2 (3.2%)	2 (4.4%)	1.000	107
Alcohol abuse	6 (5.6%)	3 (4.8%)	3 (6.7%)	0.694	107
Smoking	5 (4.7%)	1 (1.6%)	4 (8.9%)	0.159	107
COPD	8 (7.5%)	4 (6.5%)	4 (8.9%)	0.718	107
Asthma	9 (8.4%)	4 (6.5%)	5 (11.1%)	0.488	107
Hypertension	53 (49.5%)	33 (53.2%)	20 (44.4%)	0.483	107
Diabetes mellitus	34 (31.8%)	21 (33.9%)	13 (28.9%)	0.737	107
Chronic heart failure	2 (1.9%)	2 (3.2%)	0 (0.0%)	0.508	107
Chronic kidney disease	13 (12.1%)	8 (12.9%)	5 (11.1%)	1.000	107
Liver failure	1 (0.9%)	0 (0.0%)	1 (2.2%)	0.421	107
Neuromuscular disease	1 (0.9%)	1 (1.6%)	0 (0.0%)	1.000	107

BMI: Body Mass Index; COPD: Chronic Obstructive Pulmonary Disease; ICU: Intensive Care Unit; SAPS II: simplified acute physiology score 2; SOFA: Sequential Organ Failure Assessment.

**Table 2 jcm-12-02028-t002:** Comparison of respiratory and hemodynamic variables in survivors and non-survivors.

Variables	All Patients *n* = 107 Median [IQR] or Mean ± SD or *n* (%)	Non-Survivors *n* = 62 Median [IQR] or Mean ± SD or *n* (%)	Survivors *n* = 45 Median [IQR] or Mean ± SD or *n* (%)	*p*-Value	*n*
Respiratory and hemodynamics variables at baseline (Day 1)					
PEEPtot, cmH_2_O	15 [12;15]	15 [12;15]	14 [12;15]	0.696	106
Driving pressure, cmH_2_O	12 [11;16]	13 [11;16]	12 [10;15]	0.140	105
Plateau pressure, cmH_2_O	27 [25;30]	28 [26;30]	26 [25;30]	0.065	105
Crs, mL/cmH_2_O	32 [25;38]	31 [24;36]	33 [26;38]	0.254	105
EVLWi, mL/kg PBW	17 [15;22]	19 [15;22]	17 [14;21]	0.151	100
PVPI	3.6 [3.1;4.5]	4.0 [3.1;4.7]	3.4 [2.6;4.1]	**0.028**	100
CI, L/min/m^2^	2.80 [2.16;3.39]	2.70 [2.09;3.33]	2.87 [2.24;3.40]	0.503	100
CVP, cmH_2_O	11.0 [8.00;12.8]	11.5 [8.00;14.0]	10.0 [8.00;12.0]	0.462	62
PaO_2_/FiO_2_, mmHg	153 ± 79	142 ± 81	169 ± 73	0.085	104
TV, mL/kg PBW	6 ± 0.6	5.9 ± 0.5	6.0 ± 0.7	0.178	106
Maximal/minimal values of respiratory and hemodynamics variables					
PEEP max, cmH_2_0	15 [14;16]	15 [15;16]	15 [14;16]	0.238	106
DP max, cmH_2_0	17 [14;21]	18 [15;22]	16 [13;18]	**0.013**	106
Crs min, mL/cmH_2_0	24 [19;28]	24 [17;27]	26 [20;30]	**0.035**	106
EVLWi max, mL/kg PBW	22 [18;27]	24 [21;28]	19 [16;24]	**0.001**	100
PVPI max	4.7 [3.7;5.7]	5.1 [4.0;6.1]	4.1 [3.5;5.2]	**0.005**	100
CI min, L/min/m^2^	2.30 [1.94;2.71]	2.35 [1.96;2.70]	2.24 [1.85;2.75]	0.396	100
CVP max (cmH_2_O)	14.0 [11.0;17.0]	15.0 [12.0;18.0]	13.0 [11.0;16.0]	0.179	62
PaO_2_/FiO_2_ min (mmHg)	107 ± 57	89.1 ± 42	133 ± 65	**<0.001**	104
TV max, mL/kg PBW	6.3 ± 0.8	6.3 ± 0.7	6.5 ± 0.9	0.265	106
Maximal/minimal values of respiratory and hemodynamic variables during the first week					
PEEP max, cmH_2_O	15 [14;16]	15 [15;16]	15 [14;15]	0.180	102
DP max, cmH_2_O	16 [13;19]	17 [14;19]	14 [12;17]	**0.003**	102
Crs min, mL/cmH_2_0	26 [20;32]	24 [19;29]	28 [22;33]	**0.022**	102
EVLWi max, mL/kg PBW	22 [18;27]	24 [20;28]	20 [16;24]	**0.005**	97
PVPI max	4.7 [3.6;5.7]	5.0 [4,0;6.1]	4.2 [3.5;5.1]	**0.008**	97
CI min, L/min/m^2^	2.31 [1.94;2.79]	2.44 [1.99;2.71]	2.24 [1.90;2.80]	0.412	97
CVP max, cmH_2_O	13.0 [11.0;15.5]	14.0 [12.0;16.0]	12.0 [10.5;15.0]	0.103	59
PaO_2_/FiO_2_ min, mmHg	112 ± 56	98 ± 43	133 ± 65	**0.004**	100
TV max, mL/kg PBW	6.3 ± 0.8	6.2 ± 0.7	6.4 ± 0.9	0.372	102
Outcome characteristics					
Duration of MV, days	17.5 [9.00;28.8]	14.5 [8.00;26.5]	18.5 [11.0;34.2]	0.134	106
MV free days, days	0.00 [0.00;4.00]	0.00 [0.00;0.00]	9.50 [0.00;17.0]	**<0.001**	106
ICU length of stay, days	19.0 [11.0;32.0]	16.0 [9.00;29.0]	22.0 [14.0;39.5]	**0.023**	105
Hospital length of stay, days	30.0 [19.0;49.5]	25.0 [14.0;33.0]	44.5 [23.0;69.5]	**<0.001**	83

CI: cardiac index; Crs: respiratory system compliance; CVP: central veinous pressure; DP: driving pressure; EVLWi: extravascular lung water indexed; ICU: intensive care unit; IQR: interquartile range; MV: mechanical ventilation; PaO_2_/FiO_2_: ratio of the arterial partial pressure of oxygen over inspired fraction in oxygen; PBW: predicted body weight; PEEP: positive end-expiratory pressure; PVPI: pulmonary vascular permeability index; TV: tidal volume. *p*-values in bold indicate statistical significance.

**Table 3 jcm-12-02028-t003:** Correlation between respiratory and hemodynamic variables (all measurements).

	Correlation Coefficient [CI 95%]	r^2^	*p*-Value
Correlation between respiratory variables			
Pplat and Crs	−0.677 [−0.718; −0.631]	0.458	**<0.001**
Pplat and DP	0.826 [0.787; 0.841]	0.682	**<0.001**
Pplat and PEEP	0.264 [0.188; 0.338]	0.070	**<0.001**
Crs and DP	−0.825 [−0.849; −0.797]	0.681	**<0.001**
Crs and PEEP	0.275 [0.200; 0.349]	0.076	**<0.001**
DP and PEEP	−0.342 [−0.411; −0.270]	0.117	**<0.001**
Correlation between hemodynamic variables			
PVPI and EVLWi	0.77 [0.745; 0.808]	0.593	**<0.001**
Correlation between respiratory and hemodynamic variables			
Pplat and PVPI	0.059 [−0.022; 0.140]	0.004	0.139
Pplat and EVLWi	0.123 [0.043; 0.202]	0.015	**0.003**
PVPI and Crs	0.072 [−0.090; 0.153]	0.005	0.080
PVPI and DP	0.051 [−0.131; 0.035]	0.003	0.219
PVPI and PEEP	0.22 [0.141; 0.293]	0.048	**<0.001**
EVLWi and Crs	−0.003 [−0.084; 0.079]	0.000	0.951
EVLWi and DP	0.017 [−0.064; 0.098]	0.000	0.674
EVLWi and PEEP	0.203 [0.126; 0.278]	0.041	**<0.001**

CI: confidence interval; Crs: respiratory system compliance; DP: driving pressure; EVLWi: extravascular lung water indexed for ideal body weight; PEEP: positive end-expiratory pressure; PVPI: pulmonary vascular permeability index. *p*-values in bold indicate statistical significance.

## Data Availability

The datasets used and/or analyzed in the present study are available from the corresponding author on reasonable request. The data are not publicly available due to privacy.

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
