# Peer review of "Relationship of Extravascular Lung Water and Pulmonary Vascular Permeability to Respiratory Mechanics in Patients with COVID-19-Induced ARDS"

_jcm, 2023, doi:10.3390/jcm12052028_

Round 1
Reviewer 1 Report
The authors present a sound and well done retrospective analysis of correlations between respiratory variables and hemodynamic variables describing lung impairment measured by TPTD in critically ill COVID-19 patients. While the study does not find a correlation the conclusions drawn from the results are valid and well presented. I congratulate the authors on this important contribution.
Tables and figures are precise, well explained as can be grasped easily. The discussion addresses most of the implications the findings do/might have as well as most of limitations of the study.
In addition I would think of two items that could be addressed: generalizability of the results due to COVID-19 as underlying disease in which the primary insult is often a (pulmonary) vascular injury maybe leading to a "ratio" of ELWI and Crs that might differ from other forms of ARDS and which in the course of the data acquisition might be confounded by targeted treatment (steroids, anti-virals, TCZ) that became SOC during waves after the initial COVID-19 surge. Furthermore, one could address validity of the TPTD in patients with severe ACP and more than moderate tricuspid regurgitation, which can lead to false estimates of GEDVI/ITBVI and therefore ELWI. While this might not be / will probably not be adjustable in the results, it could be addressed shortly in the discussion.
On the same item, the authors could address the use of therapies in this cohort than have the potential of altering gas-exchange and respirator dynamics like NMB, proning and iNO therapy.
Author Response
Comments to the Author
The authors present a sound and well done retrospective analysis of correlations between respiratory variables and hemodynamic variables describing lung impairment measured by TPTD in critically ill COVID-19 patients. While the study does not find a correlation the conclusions drawn from the results are valid and well presented. I congratulate the authors on this important contribution.
à We thank the reviewer for these very positive comments and we are delighted that (s)he appreciated our work.
Tables and figures are precise, well explained as can be grasped easily. The discussion addresses most of the implications the findings do/might have as well as most of limitations of the study.
In addition I would think of two items that could be addressed: generalizability of the results due to COVID-19 as underlying disease in which the primary insult is often a (pulmonary) vascular injury maybe leading to a "ratio" of ELWI and Crs that might differ from other forms of ARDS and which in the course of the data acquisition might be confounded by targeted treatment (steroids, anti-virals, TCZ) that became SOC during waves after the initial COVID-19 surge.
à We thank the reviewer for these suggestions and interest in our study. Indeed, we had already mentioned the limitation regarding the generalization of our results due to the unique etiology of ARDS in our patients (‘the homogeneity of the cause of ARDS could limit the applicability of our results to other etiologies of ARDS’). We expanded the discussion by describing more specifically the impact of COVID-19 on respiratory and hemodynamic variables, and their potential consequences on our results. (‘Second, the homogeneity of the cause of ARDS could limit the generalizability of our results to other etiologies of ARDS. In a recent study comparing patients with COVID-19 induced ARDS to patients with ARDS from other causes, our team showed that ventilatory mechanics were similar but EVLW and PVPI were higher in the patients with COVID-19 ARDS [40], likely due to a higher inflammatory state. Nevertheless, this homo-geneity can also be considered as a strength of our study by overcoming the large variability that can be encountered in patients with ARDS of different etiologies..’)
We agree with the reviewer that targeted treatments such as steroids or anti-virals or tocilizumab could modulate inflammation and impact respiratory mechanics, PVPI or EVLW. We did not specifically collect this piece of information for our study. However, most of the patients included were admitted in our ICU before the NEJM publications on Tocilizumab and Dexamethasone were published, and we did not use anti-virals as a standard of care. This is now stated in the revised version of the manuscript ‘Though major publications on dexamethasone and tocilizumab modified standard of care of critically ill patients with COVID-19 during the pandemic, most of our patients were admitted before widespread use of these treatments’ (page 11, lines 263-271).
Furthermore, one could address validity of the TPTD in patients with severe ACP and more than moderate tricuspid regurgitation, which can lead to false estimates of GEDVI/ITBVI and therefore ELWI. While this might not be / will probably not be adjustable in the results, it could be addressed shortly in the discussion.
We thank the reviewer for this comment but we think that these potential confounders did not impact our results. Transpulmonary thermodilution, which detects the change in blood temperature further from the right ventricle, may be less influenced by tricuspid regurgitation than classical thermodilution with the pulmonary artery catheter, at least as suggested by an animal study (Kutter et al., Anesth Analg 2015, PMID 25742632).
In anyway, cardiac ultrasound was performed in patients of the study and none exhibited severe tricuspid regurgitation. This is now stated in the revised manuscript: ‘No right severe tricuspid regurgitation was detected in these patients through echo-cardiography.’ (page 4, lines 165-166).
On the same item, the authors could address the use of therapies in this cohort than have the potential of altering gas-exchange and respirator dynamics like NMB, proning and iNO therapy.
à We thank the reviewer for this important comment.
No patient received inhaled NO at the time of measurements. This is now stated in the manuscript (‘Inhaled nitric oxide was used in none of the patients.’ page 4, lines 166).
Some measurements were performed in prone position, as stated at page 4, lines 178-179. We fully agree that prone position could impact gas exchange and respiratory mechanics. To acknowledge this, we added in the Discussion section: ‘Twenty-two percent of our measurements were collected while patients were in prone position. Prone positioning could impact both EVLWi and respiratory mechanics on the one hand, by increasing central venous pressure and therefore impairing pulmonary edema resorption. However, prone positioning may improve ventilatory mechanics and oxygenation, decreasing hypoxemic pulmonary vasoconstriction and improving ventilatory mechanics. The final effect is still unclear. As most measurements were made in the supine position in our study, we could not specifically investigate the effect of prone positioning on the relationship between EVLWi or PVPI and respiratory mechanics.’ (page 11 lines 263-271)
Regarding the use of NMBA, patients had to be ‘passive” for allowing us to perform our measurements. Some have been paralyzed for this purpose. This is now stated in the revised version (‘For this purpose, neuromuscular blocking agents could be administered.’ page 3, lines 108-109).
Finally, we have added a statement in the discussion regarding the potential role of these treatments, acknowledging that we did not investigate it specifically ‘Fourth, we did not specifically investigate the role of prone position, inhaled nitric oxide and neuromuscular blocking agents.’(page 12, lines 338-339).
We thank again the reviewer for her/his great comments and suggestions and we have attached a marked version of our manuscript (modifications in red font) for the reviewer to be able to follow the changes made.

Reviewer 2 Report
The manuscript is of great interest. However, I have some comments:
Comments:
· “Pplat was measured during a 0.2-sec end-inspiratory occlusion” Why the authors assessed pplat after 200 ms at end-inspiratory occlusion and not after 3 seconds? This measurement is still debated.
· Isn’t the fluid balance to be considered, given the aim of the study? Please discuss
· Results: “driving pressure of 12 (11;16)”… some patients were not ventilated in a protective strategy (driving pressure > 13). see also DP max… please discuss
· I would add some more data about the different patients outcomes with respect to the hemodynamic assessments and respiratory mechanics
Author Response
Comments to the Author
“Pplat was measured during a 0.2-sec end-inspiratory occlusion” Why the authors assessed pplat after 200 ms at end-inspiratory occlusion and not after 3 seconds? This measurement is still debated.
à We thank the reviewer for this comment that will help clarify our procedure. Actually, most our Pplat measurements were assessed around 3sec during an end-inspiratory occlusion maneuver (verifying that the airway pressure was stable and without leaks). There was a confusion in our phrasing because we typically set ventilation in volume-assist control mode with an inspiratory pause around 0.2sec to continuously monitor the Pplat approximate range. This short occlusion can overestimate the actual Pplat but we use it as a warning to perform more thorough respiratory mechanics assessment when it reaches high values (Barberis L, Manno E, Guérin C. Effect of end-inspiratory pause duration on plateau pressure in mechanically ventilated patients. Intensive Care Med. 2003 Jan;29(1):130-4. doi: 10.1007/s00134-002-1568-z. Epub 2002 Dec 6. PMID: 12528034. To avoid any confusion, we modified in our methods section ‘Pplat was measured during a 3 seconds end-inspiratory occlusion and total PEEP was measured during a 3 seconds expiratory pause.’ (Page 3, lines 109-110).
Isn’t the fluid balance to be considered, given the aim of the study? Please discuss
à We thank the reviewer for this comment and we totally agree with her/him. We had already stated this point in the Limitations section (page 12, lines 402). Though fluid balance is associated with outcome in patients with ARDS, the study of the association of outcome with the different parameters measured was not part of the objective of our study. We have elaborated on this point in the discussion adding a few points in this section: ‘Finally, we did not collect fluid balance which is associated with outcomes of patients with ARDS [42]. Daily fluid balance is quite challenging to accurately collect in general in critically ill patients and this was even more difficult during the increased workload due to the COVID pandemic. Positive fluid balance could increase EVLWi and worsen respir-atory mechanics variables but it might not impact our main objective that was to assess the relationship between them.’ (pages 12-13, lines 341-346).
Results: “driving pressure of 12 (11;16)”… some patients were not ventilated in a protective strategy (driving pressure > 13). see also DP max… please discuss
à In our unit, we ventilate patients with ARDS keeping tidal volume at 6mL/kg, and trying to optimize PEEP depending on the patient potential for recruitability (using the recruitment over inflation ratio) and keeping the plateau pressure <30 cmH2O (for patients without esophageal pressure). Driving pressure is then the result of this strategy. Though we prefer patients to display a driving pressure as low as possible (and below < 13 cmH2O), as per Prof. Amato’s publications as part of a protective ventilation, we do not set ventilation to target a driving pressure value.
To our knowledge, there is no recommendation to adjust ventilation to target driving pressure and this parameter might reflect the patient’s severity (and low compliance) rather than an inappropriate ventilator setting. Based on our initial strategy, the best way to decrease driving pressure would be to lower tidal volume but this might lead to a marked increase in PCO2 with potential related complications.
This point is now discussed in the revised manuscript (‘Also, ventilatory setting were set to limit the plateau pressure, rather than the driving pressure. This may have altered our results.’page 12, lines 339-340).
I would add some more data about the different patients outcomes with respect to the hemodynamic assessments and respiratory mechanics
à We thank the reviewer for this comment. As our main goal was the assessment of the relationship between EVLW and PVPI and respiratory mechanics, we intentionally did not perform multivariable analysis to identify factors associated with the outcomes. However, we compared all measured hemodynamic and respiratory mechanics variables in survivors and non-survivors. Interestingly, we confirm that both Crs and driving pressure on the one side, and EVLWi and PVPI on the other side were worse in non-survivors than in survivors. We believe that this reinforces the message that, even if not correlated, both types of variables are important to monitor. We have added a comment on this point (‘However, our study confirms that both Crs and driving pressure on the one side, and EVLW and PVPI were worse in non-survivors than in survivors, suggesting the im-portance of these features of ARDS.’ pages 12-13, lines 287-290).
We thank again the reviewer for her/his great comments and suggestions and we have attached a marked version of our manuscript (modifications in red font) for the reviewer to be able to follow the changes made.
